

# Knowledge, perceptions, and practices of dental professionals and students regarding obturation in endodontic procedures: a nationwide cross-sectional survey

Kalyani Garde[1], Ajinkya M. Pawar[1], Anuj Bhardwaj[2], Jatin Atram[1], Suraj Arora[3], Dian Agustin Wahjuningrum[4], Maria Febritania Wahyuni Huri[4] and Dennys Kurnia[4]

[1] Conservative Dentistry and Endodontics, Nair Hospital Dental College, Mumbai, Maharashtra, India
[2] Department of Conservative Dentistry and Endodontics, College of Dental Sciences & Hospital, Rau, Indore, Madhya Pradesh, India
[3] Department of Restorative Dental Sciences, College of Dentistry, King Khalid University, Abha, Saudi Arabia
[4] Department of Conservative Dentistry, Faculty of Dental Medicine, Universitas Airlangga, Surabaya City, East Java, Indonesia

Corresponding author
Dian Agustin Wahjuningrum,
dian-agustin-w@fkg.unair.ac.id

## ABSTRACT

**Background.** The purpose of this study is to analyze dental professionals' and students' current understanding, attitudes, and usage of obturation trends in endodontic management.

**Methods.** A national cross-sectional survey of 422 dental professionals and students was carried out using a structured questionnaire The questionnaire included 34 multiple-choice questions concerning demographics, knowledge, attitudes, and practices relevant to endodontic obturation. A combination of convenience and snowball sampling was used to recruit participants. The study applied descriptive statistics and the chi-square test of proportion with a significance threshold of $p < 0.05$ and a 95% confidence interval for analysis to assess the participants' comprehension, attitudes, and use of obturation procedures.

**Results.** The study comprised 422 participants, the majority of whom were females (68.0%) with an average age of $24.55 \pm 3.31$ years. Final-year students, interns, postgraduate students, dentistry faculty, and private practitioners were among those who took part. The majority of respondents (91.2%) correctly identified the goal of root canal obturation and identified Gutta Percha and Sealer as the standard obturation material (98.8%). Obturation has a significance in root canal treatment results, according to 81.0% of respondents. 64.5% were willing to explore new obturation methods, and 72.0% said they would use newer obturation materials or equipment if it was advantageous. In routine, lateral compaction was the primary obturation technique (73.2%), with just 4.0% using a rubber dam on a regular basis. Only a small percentage (11.1%) were happy with the present efficiency and predictability of obturation procedures, whereas 58.8% noted occasional difficulties in attaining proper root canal obturation. Documentation methods were subpar, with 43.6% regularly noting obturation techniques and materials in patients' records. A sizable proportion (58.5%) underwent retreatment operations for unsuccessful obturations, highlighting

the need for improved obturation outcomes. Bioceramic sealer use and comprehension varied greatly (11.1% utilized it).

**Conclusions**. In conclusion, this study emphasizes the need of better knowledge and application of obturation trends in endodontics. Key findings include a desire for better materials, an understanding of the importance of obturation, and an openness to new ideas. The difficulties in getting optimal results underline the need of standardized education.

## INTRODUCTION

Endodontic therapies are essential for maintaining and restoring teeth that have been impacted by pulpal or periapical diseases. In endodontics, illnesses and ailments affecting the dental pulp and the tissues around it are diagnosed, treated, and prevented. It entails removing the diseased or damaged pulp, cleaning the root canal system, and then obturating the canal to seal it and stop reinfection (*Li et al., 2014*). Saving the natural tooth, which is an indispensable component for preserving oral health, function, and aesthetics, is the major objective of endodontic therapy. Patients can save expense by conserving their natural teeth and avoiding more invasive and expensive treatment choices like extractions and prosthetic replacements. Furthermore, good endodontic therapy reduces pain and inhibits the spread of infection, improving an individual's standard of life and general wellbeing (*Avila et al., 2009*).

An essential phase in endodontic therapy is obturation, which involves filling and sealing the cleaned and shaped root canal system. The aim of obturation in endodontics is to fill and seal the root canal system. The primary objectives are to avert microbial leakage, encourage periapical tissue healing, reinforce tooth structure, and provide a coronal seal to limit contamination. By accomplishing so, it anticipates to promote healing and long-term success by preventing the entry of germs and reinfection (*Sritharan, 2002*). With the advent of new endodontic breakthroughs and trends, the choice of obturation methodology, materials, and processes has changed over time. The long-term effectiveness of endodontic therapy depends on efficient obturation. Inadequate obturation can cause recurring or chronic infections, which can cause periapical pathology, treatment failure, and ultimately tooth loss (*Karamifar, Tondari & Saghiri, 2020*).

On the other hand, diligent and precise obturation procedures improve the root filling's resilience, biocompatibility, and capacity to seal, improving the treatment outcome for the affected tooth (*Badawy & Abdallah, 2022*). Additionally, clinical efficacy, chairside time, and patient comfort can all be impacted by the obturation technique and choice of material. Obturation procedures are now being optimized to retain favorable outcomes while minimizing complexity and treatment time (*Plotino, Venturi & Grande, 2018*). These developments can improve the effectiveness of endodontic practice overall, the clinical

workflow, and patient satisfaction. Practitioners' satisfaction with their chosen obturation technique is influenced by the method they use, as different techniques vary in terms of technical difficulty, efficiency, and predictability. Warm vertical compaction, although widely regarded as a highly effective technique, is more technique-sensitive and requires advanced skills and equipment, which may lead to lower satisfaction among less experienced practitioners (*AlBakhakh et al., 2022*). In contrast, single-cone obturation, often used in conjunction with bioceramic sealers, provides a simpler and more efficient workflow, leading to potentially higher satisfaction rates, particularly among general practitioners and students. Furthermore, hydraulic condensation techniques have drawn interest due to their superior sealing qualities and ease of use, which has further impacted user satisfaction levels. These techniques are preferred by practitioners who value predictability and ease of use since studies indicate that they may increase efficiency while preserving a high-quality apical and coronal seal (*AlBakhakh et al., 2022*).

Endodontics has consistently employed a variety of obturation procedures and materials, especially gutta-percha coupled with sealers employing lateral compaction or vertical condensation techniques. Recent developments have brought forward trends like thermoplasticized gutta-percha systems (such warm vertical compaction obturation) and bio ceramic-based sealers, pursuing to further improve the quality of obturation, treatment results, and streamline the procedure (*Eid et al., 2021*).

Although this research mainly focuses on current trends in obturation, it also considers older materials like silver cones and amalgam to assess awareness and their possible ongoing use in certain areas or by practitioners trained in earlier times. This approach helps trace the development of endodontic practices and reveals any ongoing dependence on outdated methods. Thus, this study aimed to obtain a comprehensive and realistic understanding of obturation practices by intentionally including participants with varying levels of clinical experience, from undergraduate students to experienced professionals. This inclusive approach mirrors the real-world dental environment, where practitioners with diverse educational backgrounds contribute to treatment planning and decision-making. Consequently, this method offers a more complete perspective on the knowledge, attitudes, and practices associated with endodontic obturation. The objective was to uncover knowledge gaps, attitudes toward emerging trends, and prevalent clinical practices in order to provide insights for enhancing education and patient outcomes. The study's null hypothesis is that there is no discernible difference between dental professionals' and students' understanding, attitudes, and practices related to obturation in endodontic procedures.

## MATERIALS & METHODS

The Institutional Ethics Committee of the College of Dental Science & Hospital (CDSH/740/2023; date of approval 01-08-2023) granted ethical permission for the present investigation. Since no personally identifying information was recorded throughout the questionnaire, participants were guaranteed secrecy and anonymity, and the informed consent waiver was granted by the Ethics committee. Participants' participation was

entirely voluntary, and they were free to leave the research at any moment. Prior to the initiation of the questionnaire address, all participants were made aware of the study's aims and acquainted with the measures taken for confidentiality and privacy. All data was acquired anonymously.

## Sample size estimation

To collect the number of participants for this study, we used a combination of convenience and snowball sampling approaches. Participants were first sourced through professional dentistry groups, educational institutions, and social media sites. Following that, these respondents were requested to share the survey link with their colleagues and peers, thereby facilitating the snowball sampling process. The snowball sampling method was chosen as a feasible way to gather a broad and representative sample of dental professionals and students. This strategy uses the initial participants' social and professional networks to reach out to others in the target population, allowing the survey to be widely disseminated. It was especially useful for connecting with people from various locations and professional groups who would otherwise be difficult to reach using standard recruitment tactics.

The snowball sampling method, while effective for reaching a wide audience, carries an inherent risk of bias due to its reliance on participants' networks. To address these limitations and ensure reliable findings, we implemented several mitigation strategies. Participant recruitment was initiated through diverse and independent channels, including academic institutions, professional dental associations, and widely used online platforms such as social media. This approach facilitated outreach to a broad spectrum of individuals and reduced the potential for sampling homogeneity. Additionally, the survey was distributed across various geographic regions and professional roles within the dental field, ensuring a balanced representation of perspectives and minimizing regional or institutional biases. To further promote honest and unbiased responses, participants were assured complete anonymity, with no personally identifiable information collected. Participation was voluntary and without incentives, reducing the risk of social desirability bias or pressure to conform to perceived norms. By combining these strategies, the study successfully leveraged the benefits of snowball sampling while mitigating its limitations, capturing a diverse and representative dataset and enhancing the reliability and generalizability of its findings. Similar bias mitigation approaches have proven effective in related research (*Marcus et al., 2017*).

We used the following formula to identify a suitable sample size for the present investigation: Sample size $n = (DEFF*Np(1-p))/((d2/Z21-\alpha/2*(N-1) +p*(1-p)))$. The formula was employed through the study's parameters, which included a population size (N) of 1,000,000, a postulated frequency of the outcome component in the population (p) of 66% +/−5%, and confidence limits (d) set at 5%. The design effect (DEFF) was set to 1 because it was not applicable to this study. This calculation yielded a sample size of 345 with a 95% confidence level. To account for potential non-responses or incomplete data, an additional 20% was added to the calculated sample size. While the 20% of 345 equates to 69 participants, the final target sample size was slightly adjusted to 422 to ensure an adequate margin beyond the calculated adjustment. This adjustment considered practical

factors such as ease of participant recruitment through snowball sampling and the need for a robust dataset to accommodate subgroup analyses. Thus, the final sample size was not strictly limited to the exact 20% attrition rate but was expanded slightly to ensure the study's reliability and representativeness across the diverse demographic groups included in the survey.

## Study population
### Inclusion criteria

Participants must be currently enrolled in a dental program (undergraduate or postgraduate) or be recent graduates from a dental school. This includes:

- Final-year dental students
- Interns in dental programs
- Postgraduate dental students
- Dental faculty members
- Licensed dental practitioners

This inclusive selection promotes a deep understanding of endodontic knowledge and practices across varying levels of expertise. By involving individuals from a wide array of educational and professional backgrounds, the study captures insights that reflect real-world clinical scenarios, where treatment decisions are shaped by both foundational education and practical experience.

## Survey instrument

The study team designed the survey instrument after conducting a thorough evaluation of the literature. The questionnaire was validated and all responses were gathered with the aid of a Google form. Potential participants (dental students and professionals) were able to access the survey through WhatsApp®, Instagram®, Facebook®, and email. Participants' names and email addresses were not obtained. The poll was conducted over a six-week period. To improve participant recruitment and retention, reminder emails and social media posts were used.

There were four major sections of the questionnaire:

- Demographics: Gender, age, and professional status (*e.g.*, final-year student, intern, postgraduate, faculty, or private practitioner) were among the basic participant data collected in this section.
- Knowledge of obturation: This section contained 10 multiple-choice questions assessing participants' understanding of obturation materials, techniques, ideal properties of obturation materials, the role of sealers, and potential complications associated with obturation failure.
- Attitudes toward obturation techniques: Ten questions in this section assessed participants' readiness to purchase advanced obturation equipment, their openness to implementing innovative materials or techniques, their opinions about retreatment cases, and their level of reliability in their obturation abilities.
- Clinical practices: There were ten questions in this section that focused on the most commonly utilized methods and supplies, difficulties encountered during obturation,

the frequency of employing rubber dams, assessment and documentation practices, and retreatment patterns in unsuccessful obturations.

## Study approach

A nation-wide sample of dental professionals and dental students was targeted for this study. The evaluation manner comprised relying on a cross-sectional survey method and sending out a questionnaire to 422 dental professionals and dental students. Particular permission was secured to use the data only for this investigation. The questionnaire included 34 questions about various areas of knowledge, attitude, and practice about Obturation in Endodontic Procedures.

The questionnaire ported 34 multiple-choice questions (MCQs), with respondents being asked to choose one or more suitable responses for each question. In addition, scope was offered for specific questions to allow for supplementary feedback as applicable. The investigation's questionnaire is referred to as the "Garde, Pawar, and Atram's (GPA) Questionnaire of Knowledge, Attitude, and Practice Regarding Trends in Obturation in Endodontic Procedures." The questionnaire is protected by copyright and has been registered with the Copyright Office of India (ROC No.: L-130888/2023, registered on July 27, 2023).

The first component of the questionnaire inquired four basic questions concerning demographic information including city, gender, age, and practice type (final year, interns, postgraduate students, postgraduate students and private practice). The questionnaire's next element included 10 questions about the objective, materials, optimal qualities, sealers, obturation techniques, and difficulties related to obturation in endodontic execution. The questionnaire's third component included ten questions about attitudes toward confidence and understanding in novel obturation techniques, as well as coping with retreatment cases in the context of obturation in endodontic treatments. The last part of the questionnaire included 10 questions about the practical elements of obturation in endodontic treatments, such as issues encountered during obturation and commonly used obturation techniques.

## Validity assessment of the questionnaire

The questionnaire's content validity was determined through an iterative review process that included endodontic subject domain experts. These specialists carefully reviewed the questionnaire to ensure that it covered all important aspects of obturation knowledge, attitudes, and practices. Their ideas were carefully included, ensuring that the instrument accurately represented the study's aims. During the pilot phase of the study, 42 individuals provided data for exploratory factor analysis (EFA) to assess construct validity. The EFA approach assessed the questionnaire's underlying structure to ensure that it accurately measured the theoretical constructs of interest. The study yielded strong factor loadings, all greater than 0.91, indicating a high level of alignment between the questionnaire items and the intended concepts. It is essential to note that the 42 people in the validation procedure were not the same as the participants in the main investigation. Before the questionnaire was finally distributed, these people were chosen one at a time to evaluate its validity and reliability. They made no contributions to the main dataset that was examined in this investigation.

### Reliability assessment of the questionnaire

The reliability over time was tested by having the same set of 42 individuals complete the questionnaire twice, with a two-week delay between administrations. This approach allows for an assessment of response consistency over time. A Pearson correlation coefficient of 0.87 was calculated, showing strong test-retest reliability. Cronbach's alpha was used to assess the questionnaire's internal consistency, which indicates how strongly the survey questions correlate and contribute to the same construct. The analysis returned a result of 0.82, indicating strong reliability and consistency across the questionnaire's components.

These steps ensured that the questionnaire was both valid and reliable, providing confidence in the quality and credibility of the data collected.

### Addressing hetrogenicity of the participants

Both professional dentists (faculty and private practitioners) and dental students (final year undergraduates, interns, and postgraduates) participated in the study. Since these groups' clinical experiences differed significantly, a subgroup analysis was conducted to evaluate each group's replies independently.

### Statistical analysis

Data collection and entry were completed using Microsoft Excel version 13, while statistical analysis was performed with IBM SPSS version 21 (IBM Corp., Armonk, NY, USA). The demographic details, along with the knowledge, attitude, and practice concerning obturation trends in endodontic procedures, were evaluated through frequency and percentage calculations. For statistical comparisons, the independent $t$-test was applied to continuous variables, and the chi-square test was used for categorical data. A subgroup analysis was conducted to compare the responses of dental practitioners and students. All statistical analyses maintained a 95% confidence interval, and $p$-values below 0.05 were deemed statistically significant.

## RESULTS

The questionnaire was completed by 422 respondents. Participants' average age was 24.55 years (SD $= 3.31$). Gender distribution showed a significant difference ($p < 0.05$), with more females (68.0%, $n = 287$) than males (32.0%, $n = 135$). The sample distribution according to the respondents is presented in Fig. 1.

### Knowledge

The majority of participants (91.2%) stated that the primary intent of root canal obturation was "filling and sealing the root canal space" ($p = 0.00$). When determining when to obturate, 60.9% of respondents took into account a number of factors, such as the lack of symptoms, adequate canal dryness, and the absence of abnormal radiography findings ($p = 0.00$). The most popular obturation material was gutta-percha with sealer (98.8%, $p = 0.00$), while 97.6% of respondents identified biocompatibility, radiopacity, and setting time as the optimal obturating material qualities ($p = 0.00$) (Fig. 2). The main use of an endodontic sealer was to fill the gaps between gutta-percha and canal walls (66.8%), with

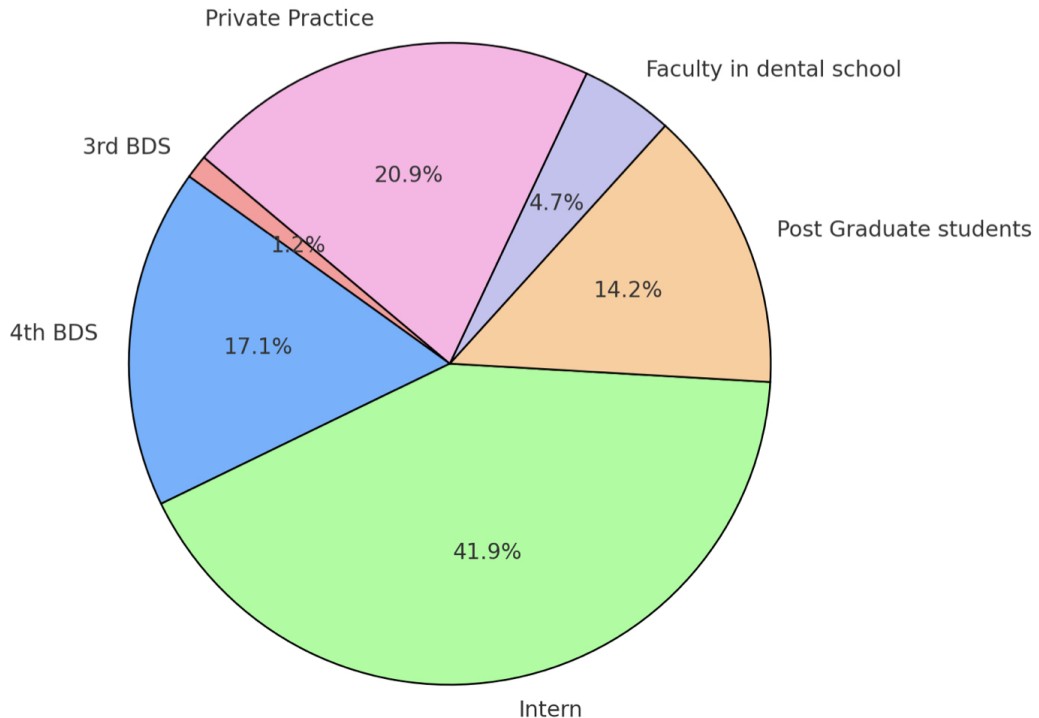

**Figure 1** Representation of the participants who completed the online questionnaire.

27% selecting several purposes ($p = 0.00$). 70.4% of participants agreed that obturation permits root canals successful by limiting bacterial recontamination, root resorption, and promoting healing ($p = 0.00$), while the majority of participants (89.3%) were aware of multiple obturation strategies ($p = 0.00$). Many people (98.1%) agreed that inadequate obturation causes a number of complications ($p = 0.00$) (Fig. 3). The beta form of gutta-percha cones was utilized the most (48.8%), with the alpha form following in second (29.4%) ($p = 0.00$). Immersion in 5.25% sodium hypochlorite for one minute was the most effective way to disinfect gutta-percha (71.6%, $p = 0.00$) (Fig. 4).

## Attitude

Particularly 10.4% (44 respondents) of the 422 participants in the study expressed great confidence in their knowledge and ability in root canal obturation. In contrast, 61.1% (258 participants) indicated a moderate level of confidence, while 1.2% (five individuals) showed no confidence at all ($p < 0.05$). The study revealed the vital importance of obturation in the overall outcome of root canal therapy, with a staggering 81.0% (342 participants) highlighting its relevance, whereas a lower number of 3.6% (15 participants) did not ($p < 0.05$). Surprisingly, 64.5% (272 participants) were enthusiastic about trying with new methods, while just 2.4% (10 participants) preferred traditional procedures above technological advancement ($p < 0.05$) (Fig. 5).

The results show that dental students and practitioners differ significantly in their confidence levels, preferred obturation methods, and material choices. Students generally

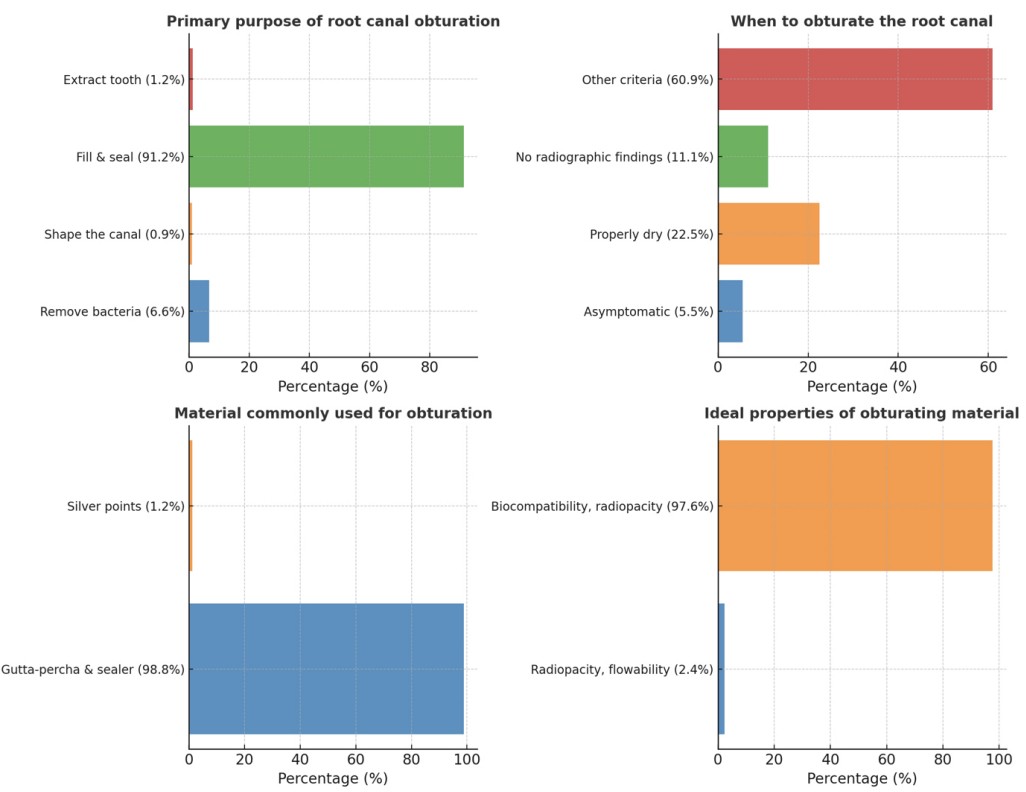

**Figure 2** Representation of the participant responses towards purpose of obturation, best time to obturate, material used for obturation, and ideal properties of obturation material.

showed moderate trust in their obturation techniques ($p = 0.041$), whereas practitioners showed higher confidence ($p = 0.032$). Students preferred the lateral compaction approach (82.1%) over practitioners (65.4%), indicating a significant difference in the adoption of obturation strategies ($p < 0.05$). Furthermore, even though rubber dam isolation has been shown to be beneficial in endodontic operations, its frequent use was still limited, with a substantially lower percentage of students (2.6%) using it consistently than practitioners (7.3%) ($p = 0.018$). Additionally, the study discovered a noteworthy discrepancy in the uptake of bioceramic sealers, with practitioners showing a stronger propensity to employ them (18.5%) than students (6.4%) ($p = 0.027$). In order to close the confidence and best practices gap between students and seasoned clinicians, these findings highlight the necessity of improved clinical training and education (Table 1).

Additional data regarding educational endeavors revealed that 38.2% (161 those who participated) occasionally attended Continuing Dental Education (CDE) Courses related to RC Obturations, 31.5% (133 participants) rarely took part and 9.2% (39 participants) never participated in any relevant course or workshop ($p < 0.05$). Further, 72.0% (304 participants) indicated an intention to implement newer obturation materials or equipment if demonstrated advantageous to their practice ($p < 0.05$) (Fig. 6). The study also investigated participants' viewpoints on obturation method developments, patient

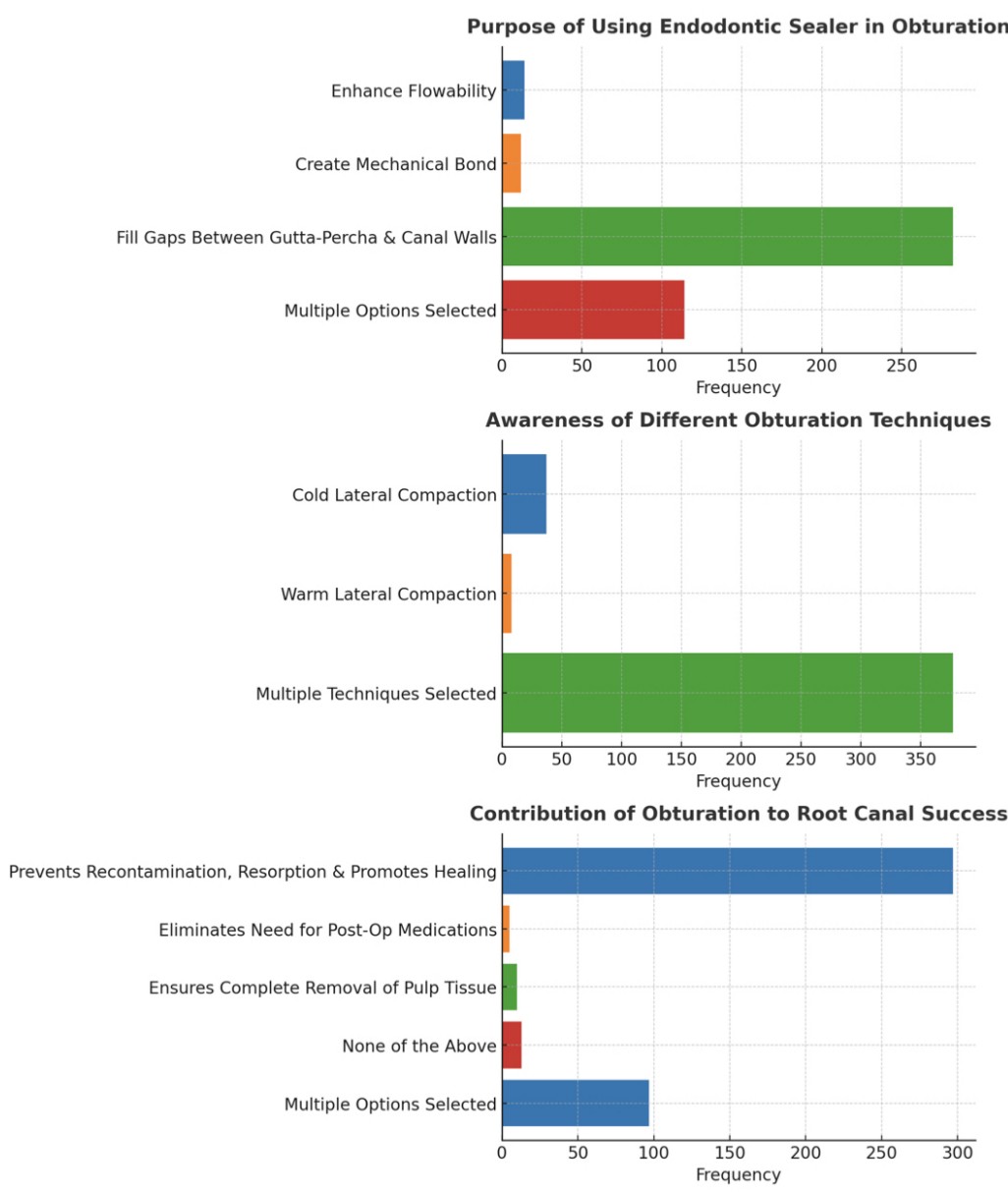

**Figure 3 Representation of the participant responses towards purpose of sealers, different techniques, and contribution of obturation.**

education, and their level of satisfaction with existing obturation materials, offering an exhaustive overview of their attitudes and beliefs in the domain of root canal therapy (Fig. 7).

## Practices

A substantial proportion of the 422 participants (73.2%) preferred employing the lateral compaction technique during obturation procedures, demonstrating its widespread use and preference ($p < 0.05$). On the other hand, a mere 17 individuals (4.0%) chose to utilize

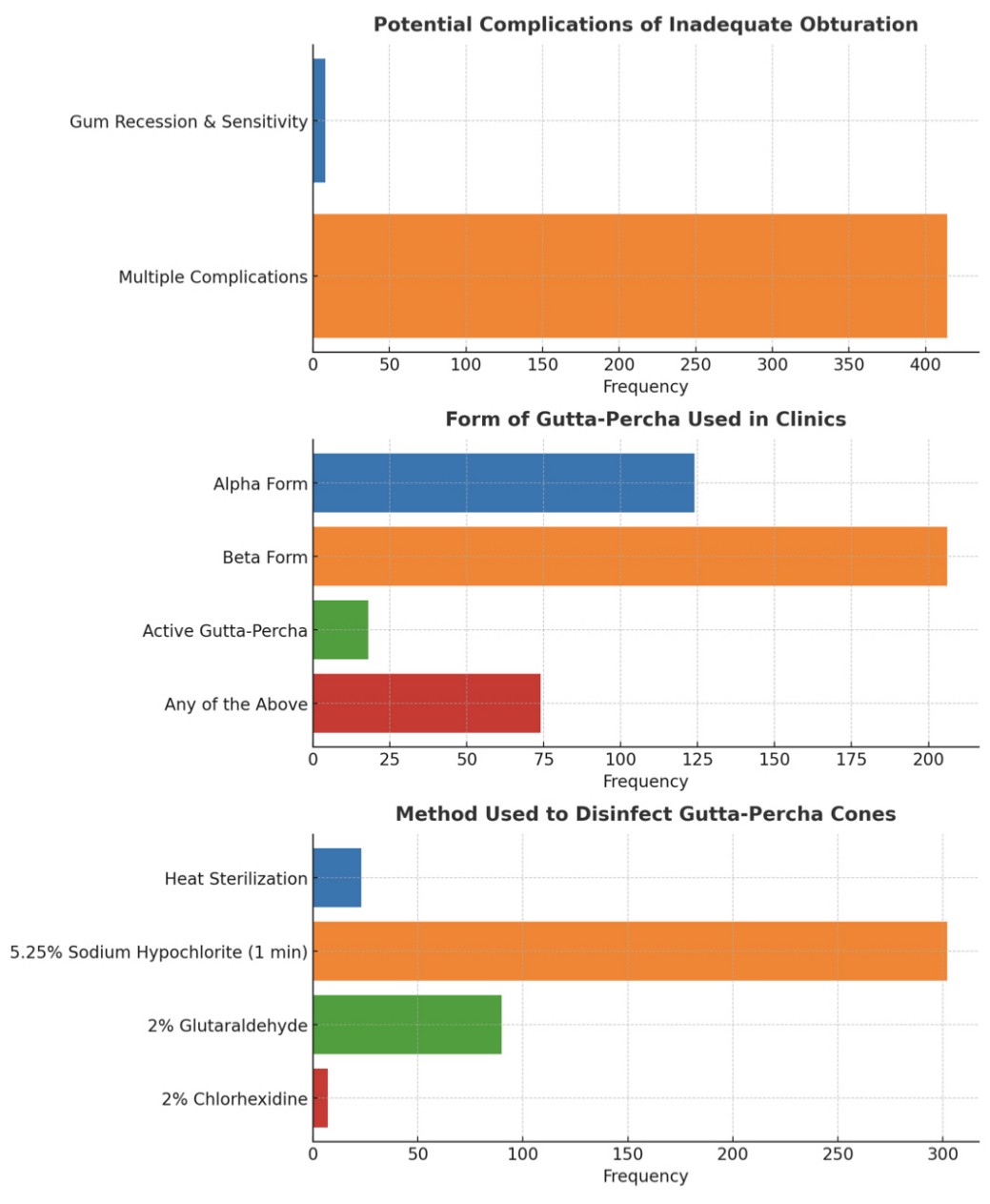

**Figure 4** Representation of the participant responses towards potential complications, form of gutta-percha, and methods of disinfecting gutta-percha during obturation.

a rubber dam on a regular basis. Furthermore, 222 individuals (52.6%) used the rubber dam only sometimes for certain circumstances, whereas 127 participants (30.1%) never used it in their obturation practices, demonstrating a difference in its use ($p < 0.05$). In regard to obturation comfort, 248 individuals (58.8%) reported occasional difficulty in obtaining adequate root canal obturation, whereas 90 participants (21.3%) infrequently noticed such difficulties ($p < 0.05$). Unexpectedly, only 47 people (11.1%) were satisfied
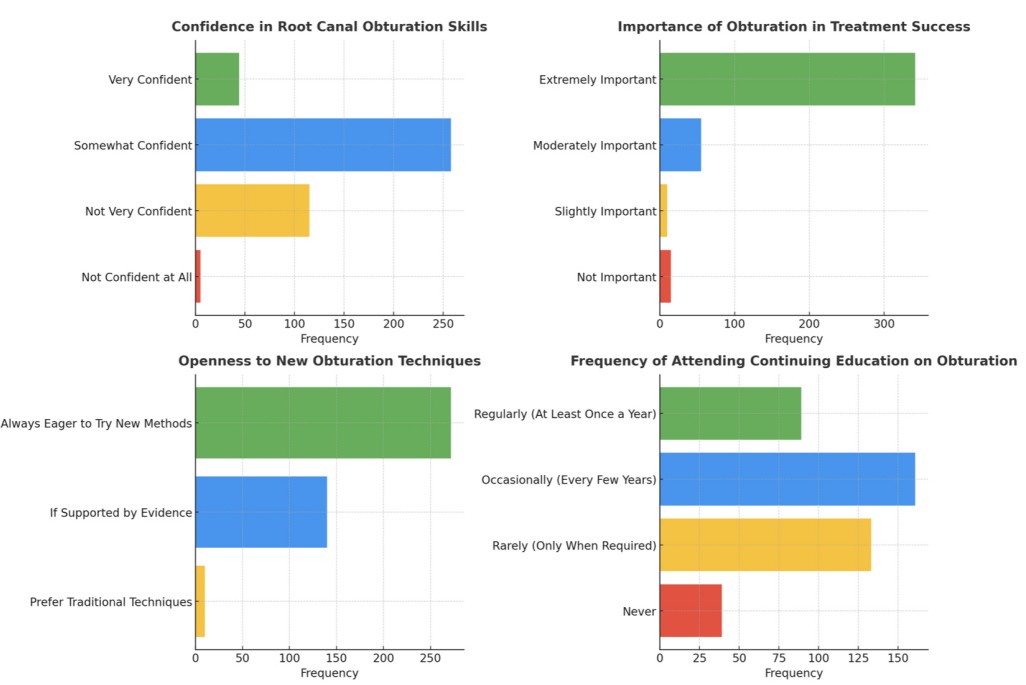

**Figure 5** Representation of the participant responses towards their confidence, their view on importance, and willingness to adopt new techniques for obturation.

**Table 1 Comparison of confidence levels, preferred techniques, and material choices between students and practitioners.**

| Variable | Students (%) | Practitioners (%) | p value | Statistical test |
|---|---|---|---|---|
| **Confidence in obturation techniques** | Moderate | Higher | 0.032 | Independent *t*-test |
| **Preferred obturation method: lateral compaction** | 82.1% | 65.4% | <0.05 | Chi-square test |
| **Frequent use of rubber dam** | 2.6% | 7.3% | 0.018 | Chi-square test |
| **Use of bioceramic sealers** | 6.4% | 18.5% | 0.027 | Chi-square test |

with the existing efficiency and predictability of obturation processes, indicating space for innovation and development in the sector ($p < 0.05$) (Fig. 8).

Furthermore, 260 volunteers (61.6%) regularly rated the quality of obturation by radiography or other imaging techniques after each obturation treatment, demonstrating a commitment to quality assessment ($p < 0.05$). In the realm of documentation, 184 participants (43.6%) recorded and documented obturation techniques and materials in patients' records frequently but not regularly, indicating a limited adherence to complete record-keeping practices ($p < 0.05$). A considerable majority of participants, 247 individuals (58.5%), completed retreatment procedures for failed root canal obturations, highlighting the need for better obturation outcomes and the necessity of retreatment ($p < 0.05$).

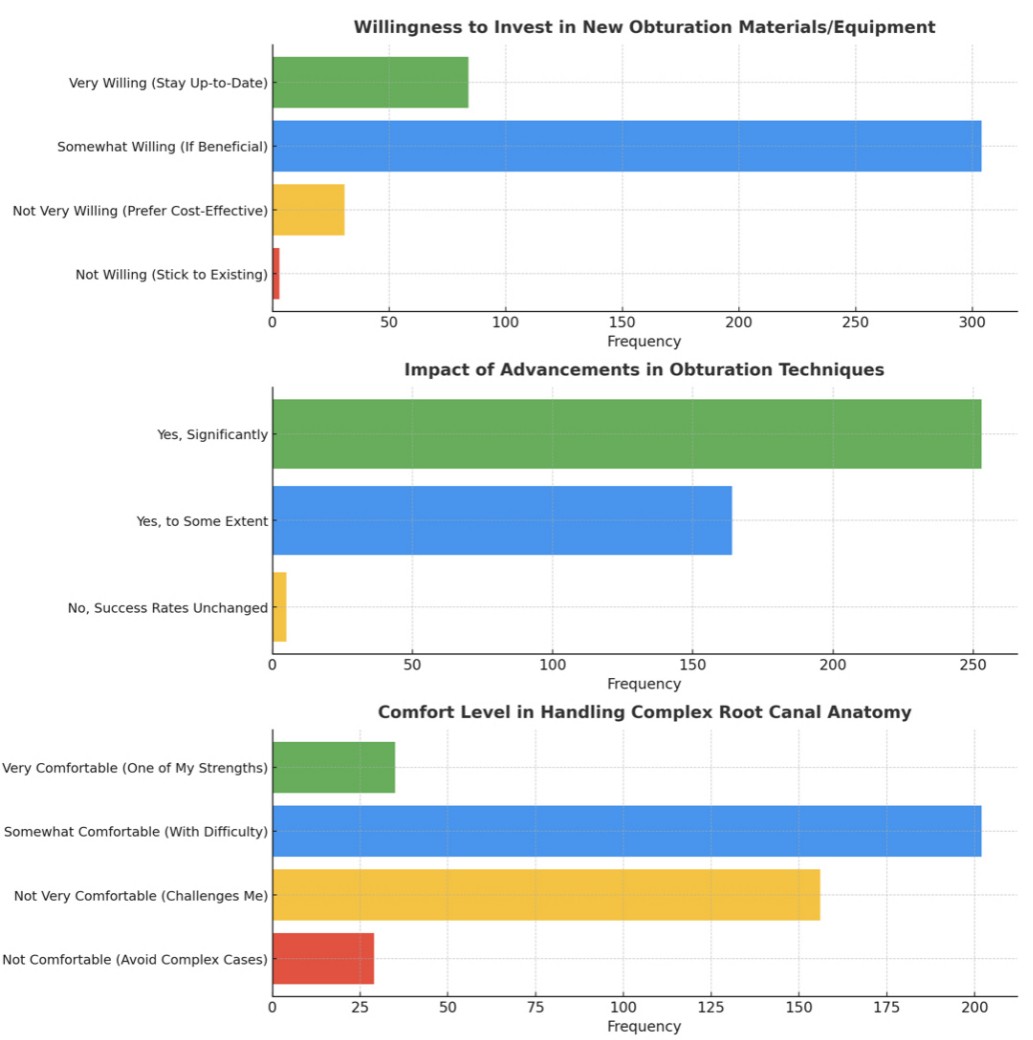

**Figure 6** Representation of the participant responses towards attending continuing dental education workshops, willingness to invest in new armamentarium, and belief that newer armamentarium is better or no for obturation.

Furthermore, only 47 individuals (11.1%) used bioceramic sealer, whereas a significantly larger group of 173 participants (41.0%) did not, indicating a disparity in acceptance and understanding of this specific sealer ($p < 0.05$) (Fig. 9).

Other than that, participant involvement with feedback from patients differed significantly, with 202 participants (47.9%) obtaining feedback on the patient experience with root canal obturation on a frequent schedule, while a small percentage of five participants (1.2%) never sought feedback and did not consider it important ($p < 0.05$). In the context of upgrading knowledge and abilities, 131 participants (31.0%) actively attempted to improve their understanding of modern root canal obturation techniques, while 246 individuals (58.3%) updated their knowledge as new techniques surfaced on occasion. Surprisingly, 45 participants (10.7%) relied simply on their current knowledge

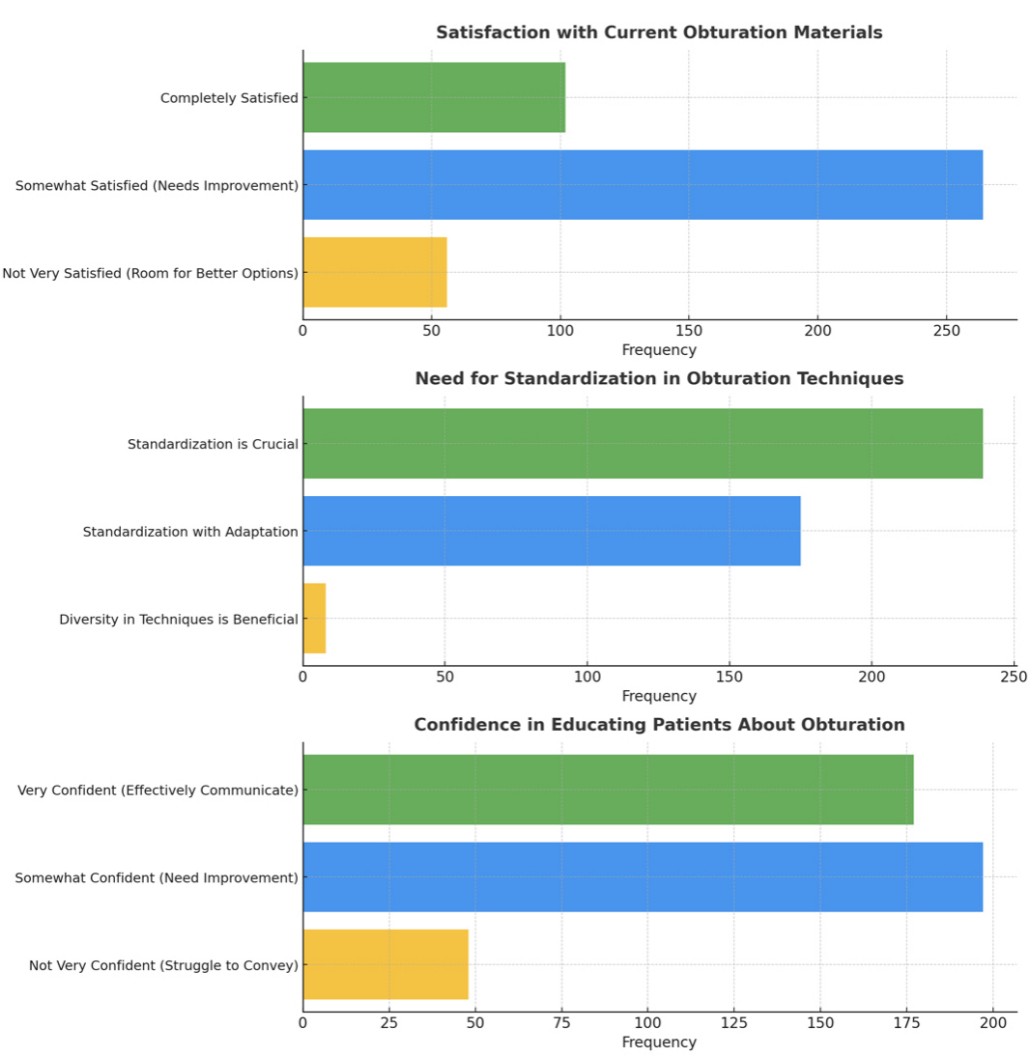

**Figure 7** Representation of the participant responses towards handling complex cases, current obturating materials, need for standardization, and confidence about educating the patients towards obturation.

and skills, highlighting the participant pool's diverse approaches to skill growth ($p < 0.05$) (Fig. 10).

A stratified analysis was performed to evaluate how different participant groups affect endodontic knowledge, attitudes, and practices.

**Knowledge-based responses:** Most respondents (65%) identified preventing reinfection as the main goal of obturation. However, among third-year Bachelor of Dental Surgery (BDS) students, 55% prioritized completely sealing the canal, highlighting a difference in understanding at various academic levels. A significant portion (70%) of participants believed obturation should be done when the canal is dry and asymptomatic. In contrast, 60% of faculty members preferred immediate obturation after shaping and cleaning, reflecting differing clinical decision-making approaches. Gutta-percha was unanimously

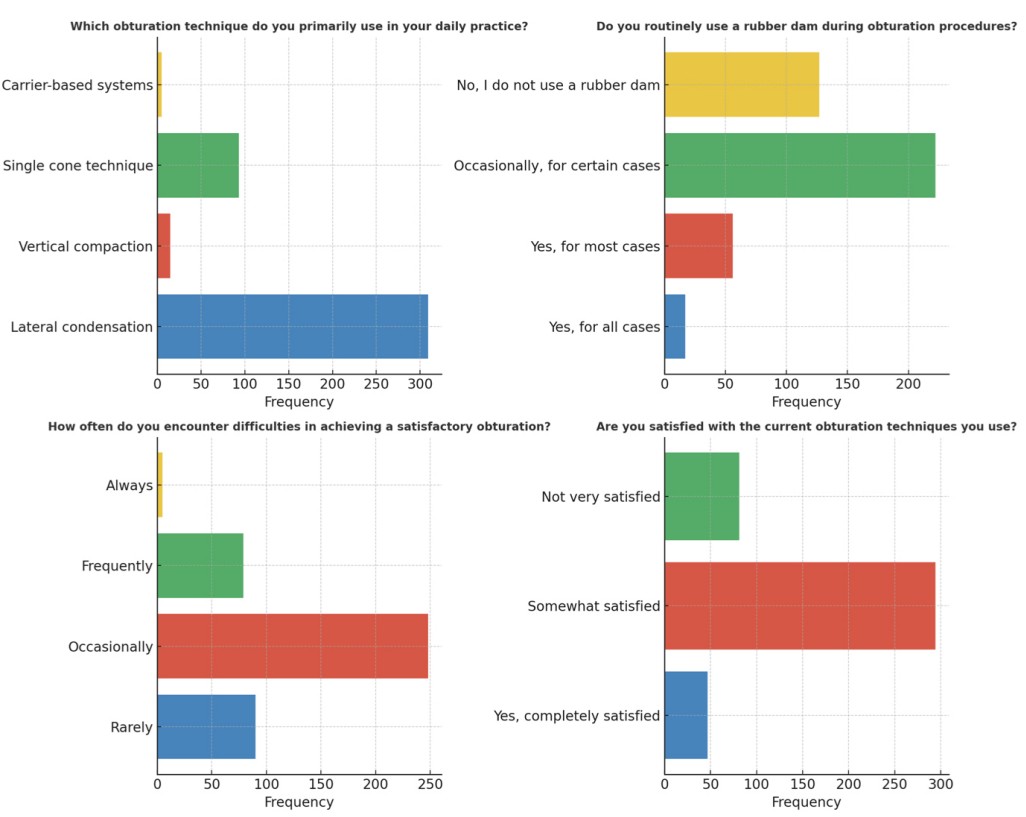

**Figure 8** Representation of the participant responses towards obturation techniques, rubber dam isolation, difficulties and satisfaction with the current obturating techniques.

chosen as the material of choice by all participant categories (100%). A vast majority (85%) emphasized biocompatibility and effective sealing ability as the most crucial properties of an ideal obturation material. All participants (100%) agreed that the primary function of an endodontic sealer is to fill voids and enhance the seal, ensuring successful obturation. Awareness levels varied among groups, with 90% of faculty and postgraduate students showing a strong understanding of potential complications, while only 60% of third- and fourth-year BDS students reported similar awareness. Sodium hypochlorite was correctly identified as the most effective root canal disinfectant by 80% of postgraduates and faculty members. However, uncertainty was noted among undergraduates, with 50% unable to confidently identify the preferred disinfectant.

**Attitude-based responses:** Faculty members exhibited the greatest self-assurance, with 75% rating themselves as "highly confident." Conversely, students showed less confidence, with 65% reporting "moderate" or "low" confidence levels. Every participant (100%) agreed that obturation is crucial for the success of root canal procedures. Although 80% of respondents were "very open" to embracing new techniques and materials, faculty members and third-year BDS students were somewhat more cautious, with 55% expressing "moderate openness." Most participants (70%) attended such courses occasionally, while 65% of faculty and postgraduates attended more frequently, indicating a dedication

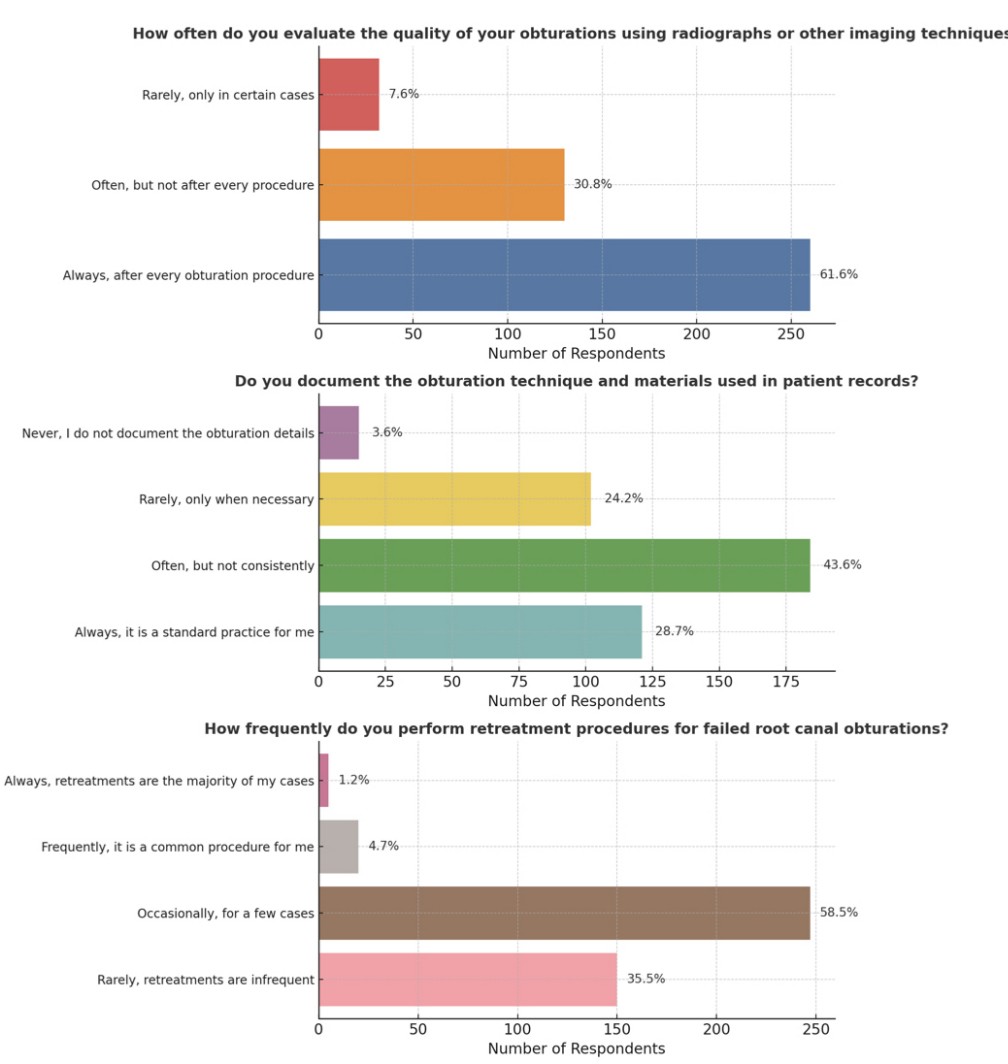

**Figure 9** Representation of the participant responses towards evaluation, documentation, retreatment and usage of bioceramic sealers for obturation.

to lifelong learning. Overall, 60% of participants were "somewhat willing" to invest in new obturation materials, with postgraduates and private practitioners showing a higher willingness at 75%. Higher confidence levels (80%) were observed among faculty and postgraduates, whereas only 40% of students felt confident in managing complex anatomical variations. Satisfaction levels varied, with 50% of professionals satisfied with current materials. However, 70% of students and interns felt that improvements were needed.

**Practice-based responses:** Among interns, postgraduates, and private practitioners, 80% primarily employed the "cold lateral compaction" technique. Conversely, 75% of faculty members and third-year BDS students favored the "warm vertical compaction" method. Fourth-year BDS students displayed a mixed preference, with 55% opting for "cold lateral compaction." Regular use of rubber dams was reported by 70% of interns, while

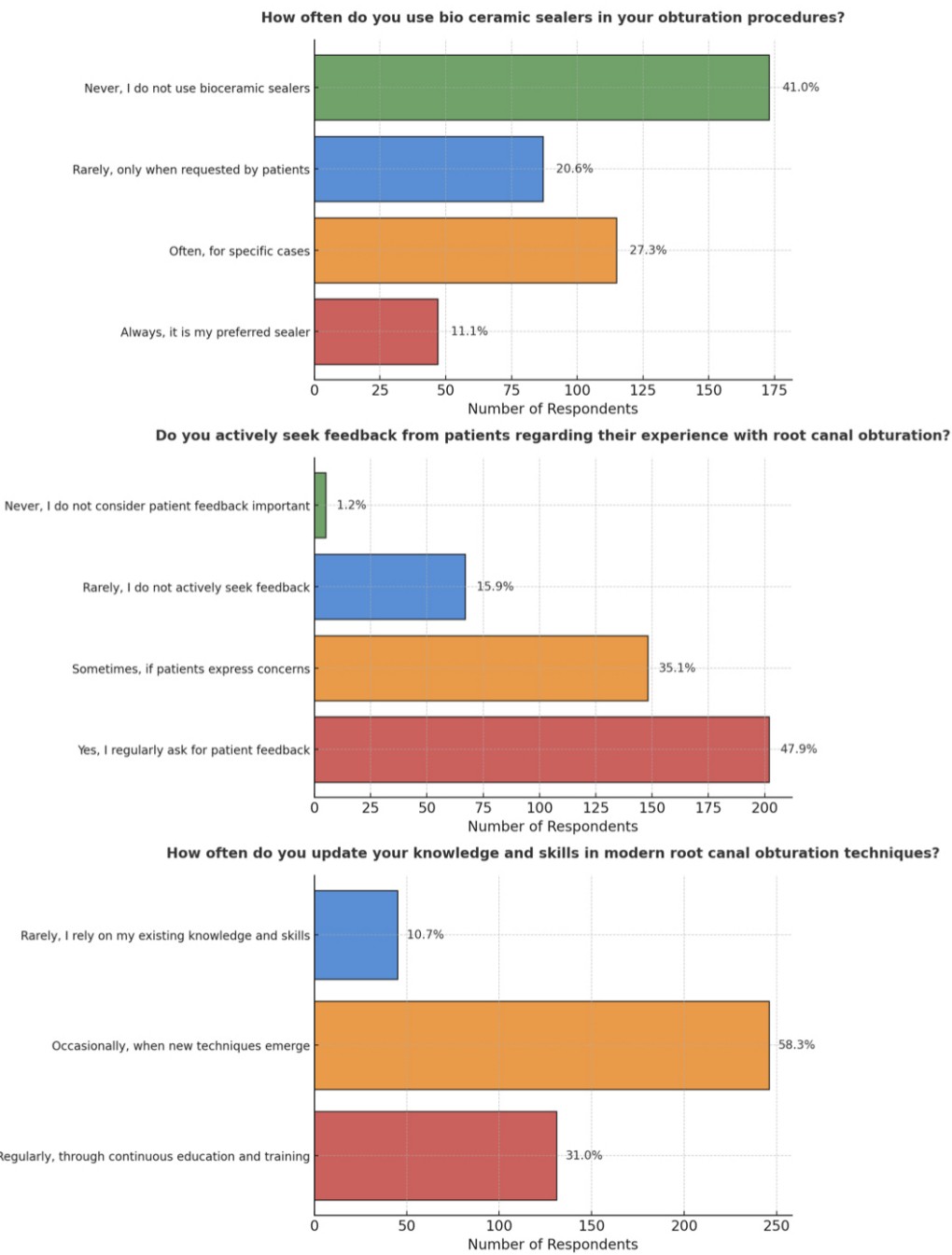

**Figure 10** Representation of the participant responses towards active feedback and updating of their knowledge with respect to obturation.

faculty and professionals indicated moderate use, with 65% using them "occasionally." Students reported a lower frequency, with 60% using them "rarely" or "occasionally." A significant 85% of participants consistently used radiographs to confirm obturation quality. Retreatment was occasionally performed by 70% of faculty members, interns, and

postgraduates, whereas private practitioners reported a lower frequency, with 65% "rarely" engaging in such procedures. Frequent updates were noted by 80% of faculty members and postgraduates, while 60% of students and interns reported "occasional" updates. There was strong support for standardization among 85% of faculty members and postgraduates, but only 50% of students and interns were certain about its necessity. A proactive approach to patient education was adopted by 90% of professionals, while 60% of students and interns reported "occasional" efforts in this area.

## DISCUSSION

The study that intended to evaluate dental students' and professionals' knowledge, attitudes, and actions on obturation trends in relation to root canal procedures in India. The key objective of the research was to assess how the students and clinicians cope with obturation, a crucial component of root canal therapy that involves sealing up the cleaned and formed root canal with the goal to avoid reinfection and preserve tooth integrity. Students in the first to final year of undergraduate study, interns, postgraduate endodontics students, postgraduate students in other dental specialties, dental instructors from all over India, and private dental practitioners are among the dental professionals that took part in the study. The individuals involved were chosen because to their core expertise required for effective execution and their practical experience providing root canal treatments. In India, general dentists and endodontists are equally skilled in performing root canal operations. While endodontists are recognized as specialists in this field, general dentists who have access to the proper study resources and practical experience can develop skills on par with those who have had formal training. The primary emphasis lies on the significance of continuing education, sustaining contemporary with technical improvements, and staying knowledgeable about the most recent research to deliver high-quality care to patients.

The study intentionally referenced older obturation materials like silver cones and amalgam, alongside evaluating current practices. This approach aimed to capture any remaining knowledge, continued use, or perceptions among practitioners trained with older methods or working in resource-constrained environments. Although these materials are mostly outdated in modern endodontics, their inclusion helps identify educational gaps, outdated practices, and areas needing ongoing professional development. Our analysis highlights that contemporary materials, such as gutta-percha and bioceramic sealers, are still predominantly used in clinical settings, as confirmed by the responses. The historical materials were not the main focus but were included to contextualize changing trends and ensure a thorough understanding of endodontic obturation knowledge across different experience levels.

Engaging participants with a range of experience levels in research studies provides a robust framework for evaluating educational gaps and variations in clinical practice. This strategy allows researchers to monitor how theoretical knowledge is developed and applied at different stages of professional growth (*Christidis et al., 2024*). Bringing together people with diverse educational and professional experiences enabled a comprehensive evaluation of current trends, challenges, and knowledge gaps. This diversity reflects the realities of everyday dental practice and helps pinpoint areas where improvements in

training, continuous education, or clinical methods might be necessary. Instead of skewing the data, this variety enhanced the findings by providing insights across the entire spectrum of endodontic practice.

The findings showed that students and practitioners differed statistically significantly in a number of areas, such as confidence levels, preferred obturation methods, and material selections. Like for instance, practitioners were more inclined to use contemporary obturation materials and techniques and showed higher levels of confidence than students. The study also emphasized differences in clinical practice patterns, such as being the case that students use lateral compaction more frequently than experienced clinicians, who employ a wider range of procedures. These results imply that attitudes and behaviors around endodontic obturation are significantly influenced by professional experience and clinical exposure. The null hypothesis is rejected in light of the observed differences, suggesting that knowledge, attitudes, and practices related to obturation are greatly influenced by educational background, clinical experience, and exposure to continuing education. The results of the research are consistent with established hypotheses and volume of knowledge in the discipline of endodontics. It emphasizes the percentage of participants correctly recognized the main goal of root canal obturation, which is to fill and seal the root canal space (*Patel, 2016*; *Somani et al., 2019*; *Kikly et al., 2020*).

Regarding the time of obturation, clinical practice underwent a significant change (*Tabassum & Khan, 2016*). In earlier times, therapy was initiated determined by the presence or absence of certain symptoms, with treatment selections mostly based on symptoms (*Bergenholtz, 2016*; *Pontoriero et al., 2021*). Our research indicates a substantial shift in this strategy, nevertheless. Nowadays, the majority of people think about factors other than symptoms when figuring out when obturation is appropriate. This movement reflects a change in endodontic treatment, indicating a tendency toward a more thorough review of clinical and radiographic features before moving forward with obturation. The need of a comprehensive examination is emphasized by the increasing knowledge when deciding whether to obturate a root canal treatment.

One key factor influencing the responses in this study is the participants' experience level. Private practitioners, due to their greater clinical exposure, may exhibit higher confidence in obturation techniques and encounter fewer challenges than third year students or interns. Previous studies have demonstrated that confidence in performing endodontic procedures increases with experience, as practitioners refine their skills and develop greater familiarity with various obturation methods (*Madfa et al., 2025*). Furthermore, practitioners' judgments of success rates and difficulty may be greatly impacted by the obturation technique they choose. Because of its simplicity and convenience of usage, lateral compaction is still the most popular technique and is favored by both general practitioners and students. On the other hand, more sophisticated methods like thermoplasticized gutta-percha or warm vertical compaction call for more training and specialized tools, which could explain why people report difficulties and differing degrees of confidence (*Bender, Ocak & Uzunoğlu Özyürek, 2024*). While people who are less experienced may view these methods as more technically complex, practitioners who are skilled in them may feel more assured about attaining ideal obturation.

 

Added to that, the adoption of gutta-percha and sealer as obturation materials demonstrates that there is widespread concurrence on their efficacy and therapeutic acceptability. Moisture resistance, bactericidal capabilities, and non-staining characteristics are acknowledged aspects of an ideal obturating material that are congruent with the desirable traits mentioned in current literature (*Tomson, Polycarpou & Tomson, 2014*; *Vishwanath & Rao, 2019*).

This demonstrates a clear agreement between practitioners' opinions and defined material selection guidelines. Recognizing the significance of the endodontic sealer in filling spaces between gutta-percha and canal walls corresponds with existing information, confirming the necessity of creating a tight seal inside the root canal system (*Schwartz, 2006*). Overall, our findings confirm current knowledge while also pointing to a changing environment in endodontics, where practitioners are blending established concepts with evolving viewpoints to enhance therapeutic treatments and improve patient outcomes.

When the results of the present study are compared to the existing literature, it throws illumination on the spectrum of practitioners' views and practices in endodontic obturation. According to the findings of the study, a sizable proportion of participants (61.1%) reported moderate confidence rather than high confidence (10.4%) in their knowledge and abilities connected to root canal obturation. This is consistent with prior study, which suggests a requirement for continued education and skill building to boost practitioner confidence, stressing a continuous learning approach in the field (*Aulakh, 2022*). One remarkable finding was the overwhelming awareness (81.0%) of obturation's imperative role in root canal treatment effectiveness. This is consistent with known information (*Ng, Mann & Gulabivala, 2011*), which emphasizes obturation as a vital component in producing beneficial results. Obturation's significance underscores practitioners' knowledge of its function in avoiding reinfection and encouraging healing.

Additionally, the study discovered a significant proclivity towards using innovative obturation techniques and technologies. A sizable majority (64.5%) showed eagerness to test new treatments, demonstrating a progressive mentality among the dental industry. This is consistent with the field's developing character, as practitioners are constantly looking for ways to improve the efficiency and efficacy of root canal obturation (*Kishen et al., 2016*). This demonstrates a proactive attitude to remaining current on the newest trends and procedures in obturation, which aligns with the need of continuing education in endodontics (*Madarati & Habib, 2018*). This is essential in a field characterized by the continuous introduction of novel materials and technologies, reflecting a proactive strategy aimed at enhancing treatment outcomes (*Zitzmann et al., 2010*; *Kishore et al., 2014*).

Lateral compaction was preferred as the principal obturation technique in this study (73.2%), which is in line with findings from other studies that emphasize its dependability, affordability, and user friendliness (*Lee et al., 2009*; *Bhandi et al., 2021*). The decreased use of alternative methods including single-cone obturation, warm vertical compaction, and thermoplasticized gutta-percha systems can be attributed by a number of causes. In order to reinforce its common clinical use, the majority of dental students and professionals undergo substantial training in lateral compaction during their undergraduate studies. Additionally, some practitioners may not be able to afford the specialist equipment needed
for methods like warm vertical compaction, such as obturation guns and gutta-percha warmers, especially those working in environments with limited resources (*Eid et al., 2021*).

The challenging process of learning and technique sensitivity of alternative procedures are also important factors preventing their widespread acceptance. Greater precision is essential when employing warm vertical compaction and thermoplasticized obturation procedures, and inadequate control might result in overextending of the material or insufficient sealing (*Gomes, Aveiro & Kishen, 2023*). Furthermore, despite the benefits of newer methods like single-cone obturation with bioceramic sealers, such as decreased technique sensitivity and enhanced sealing ability, their uptake is still low (11.1%) because of formal training exposure and worries about long-term clinical performance (*AL-Haddad & Che Ab Aziz, 2016*). In order to promote broader use of advanced obturation procedures in clinical practice, the current study emphasizes the necessity of placing more emphasis on these techniques in postgraduate curricula and continuing dental education (CDE) programs.

The challenges involved with obturation are further highlighted by the high incidence of retreatment procedures (58.5%) seen in this study. Microleakage caused by poor obturation quality results in chronic periapical infections that require reintervention (*Tabassum & Khan, 2016*). Retreatment has a substantial effect on patient outcomes since it prolongs treatment, raises costs, and, in certain situations, lowers the impacted tooth's long-term prognosis. According to studies, the capacity to remove prior filling material, thoroughly clean the canal system, and produce a better seal during the secondary obturation phase are all critical to retreatment success rates (*Gomes, Aveiro & Kishen, 2023*). To reduce the chance of failure and enhance long-term treatment results, practitioners should give special attention to thorough cleaning and shaping, accurate obturation methods, and long-lasting coronal restorations (*Azarpazhooh et al., 2022*).

The significantly low frequent use of a rubber dam (4.0%) throughout obturation, on the other hand, raises concerns, since its constant application is critical for effective infection management and better obturation results (*Bhandi et al., 2021*). The uncommon use of a rubber dam, as well as reported difficulties in getting proper obturation (58.8%), underscore the need to stress and reinforce optimal practices, with the ultimate goal of improving accuracy and efficacy in obturation procedures (*Nasser, 2021*). Only 11.1% of participants were unhappy with their current efficiency and predictability of obturation processes, indicating that there is opportunity for development and innovation in the industry. This shows a discrepancy between existing obturation procedures and the intended degree of efficacy and predictability, highlighting the need for research and development to address these problems. The disparity in bioceramic sealer acceptability (11.1%) instead of the substantial percentage of people not utilizing it (41.0%) highlights the need for increased education and knowledge regarding emerging obturation materials like Bioceramic Sealer.

Furthermore, our results about the limited application of bioceramic sealers are consistent with previous research, indicating a delay in practitioners' adoption of new materials (*Dong & Xu, 2023*). Due to their expensive cost, lack of familiarity, and limited access to hands on training, bioceramic sealers are not widely used, despite their excellent

sealing ability, biocompatibility, and antibacterial qualities (*AL-Haddad & Che Ab Aziz, 2016*). Given concerns about cost-effectiveness and long-term therapeutic performance, many practitioners favor conventional epoxy-based sealers (*Jain et al., 2019*). The selection of obturation materials by practitioners is greatly influenced by economic factors in addition to educational disparities. Because they are still less expensive, traditional sealers are a more sensible choice, particularly in environments with limited resources (*Aminoshariae, Kulild & Nagendrababu, 2021*). This implies that worries about accessibility and cost may be just as much of the reason for the hesitancy to use bioceramic sealers as a lack of knowledge. Targeted educational initiatives and financial incentives might be required to help integrate bioceramic sealers into standard practice, as 41.0% of participants said they did not use them despite their potential benefits.

Overcoming possible misunderstandings and increasing knowledge of such materials' advantages and applications is critical for their effective incorporation into clinical practice (*Parirokh & Torabinejad, 2010*; *AL-Haddad & Che Ab Aziz, 2016*). Furthermore, the active pursuit of patient suggestions and attempts to improve awareness of current obturation procedures demonstrate a proactive, patient-centric attitude as well as a readiness to adapt to emerging practices (*Bansal & Jain, 2020*; *Manasa et al., 2023*). Encouragingly, a sizable percentage of practitioners are receptive to continual learning and skill improvement, which is critical for keeping up with the most recent breakthroughs in endodontic obturation.

The research on endodontic obturation procedures has important significance for developing and refining your treatment approach. Our findings highlight the critical need of using a rubber dam consistently during obturation procedures to achieve infection control and aseptic circumstances. It is critical to prioritize training and continuing education programs in order to solve the observed difficulties in attaining proper root canal obturation and improve overall practitioner satisfaction. It is suggested to delve into and familiarize oneself with cutting-edge obturation materials, such as Bioceramic Sealers, to facilitate their integration and widespread use in clinical settings (*Dong & Xu, 2023*). Moreover, promoting patient-centric care by actively seeking and comprehending feedback from patients may dramatically improve treatment quality. Finally, we emphasize the significance of thorough record-keeping techniques for complete patient care and good treatment results.

An essential part of evaluating the quality of dental care is patient feedback. Patients' opinions on post treatment symptoms, procedural discomfort, and general satisfaction can serve as an indirect indicator of the effectiveness of obturation techniques, even though input is usually requested regarding the overall results of root canal therapy rather than particularly regarding obturation. According to earlier studies, patient reported experiences can improve patient education and treatment protocols, which are crucial for establishing reasonable expectations and guaranteeing adherence to post treatment care (*Wong, Mavondo & Fisher, 2020*; *Han & Liang, 2024*). However, we recognize that our study did not correlate patient feedback with specific obturation techniques.

A significant limitation of this study is the questionnaire's failure to evaluate confidence levels linked to specific obturation techniques. Practitioners' confidence in executing obturation procedures is likely to vary depending on whether they use lateral compaction,

warm vertical compaction, thermoplasticized methods, or the newer single-cone techniques with bioceramic sealers. Without technique-specific data, the differences in responses might be due to the inherent complexity or the practitioner's familiarity with the technique used. This factor should be taken into account when interpreting the findings related to confidence. We recommend that future research explore the relationship between obturation techniques and confidence levels to gain a better understanding of how training and technique complexity affect practitioners' self-confidence (*Madfa et al., 2025*). One of the study's limitations is that answers to some questions about attitude and practice were not immediately connected to the methods and resources participants employed for obturation. This aspect should have been taken into consideration because practitioners' use of lateral compaction, warm vertical compaction, or thermoplasticized techniques may have an impact on their confidence levels, willingness to adopt new techniques, and routine clinical practices (*Kucuk, Ratakonda & Piasecki, 2024*).

The inclusion of easier to reach or driven individuals may have been skewed by convenience and snowball sampling procedures, affecting the generalizability of the findings to the larger community of dental professionals and students. Furthermore, response bias might be a constraint, as individuals with a vested interest or strong opinion may have been more likely to engage, potentially overrepresenting certain opinions. When relying on self-reported data from a questionnaire, memory bias and answer accuracy concerns may arise, possibly reducing the accuracy and depth of the received information when compared to direct observations or clinical examinations. The majority of the participants in the study were dental students, interns, and postgraduate students, with a lesser number of practicing dentists and faculty members. The variation in confidence levels highlighted in the survey could be linked to the different obturation techniques preferred by the participants. The lack of separate analysis for each technique might have contributed to the variability in the results. Conducting future research that assesses confidence levels for each specific technique could offer more detailed insights.

## CONCLUSIONS

In conclusion, this study highlights the critical need for enhanced knowledge and application of current trends in obturation among dental students and professionals. The findings reveal a strong understanding of the fundamental objectives of root canal obturation, with a significant majority recognizing the importance of effective sealing to prevent reinfection. Participants demonstrated familiarity with traditional materials like gutta-percha and expressed a clear interest in exploring newer obturation techniques and materials, indicating an openness to advancements in endodontic practice.

## ACKNOWLEDGEMENTS

The questionnaire employed in the current study is registered with the Copyright Registry of India, with the title "Garde, Pawar, and Atram's (GPA) Questionnaire of Knowledge, Attitude, and Practice Regarding Trends in Obturation in Endodontic Procedures" and ROC No.: L-130888/2023, registered on July 27, 2023.

### Funding

This work was supported by the Deanship of Scientific Research and Graduate Studies at King Khalid University in Abha, Saudi Arabia, through the Large Research Group initiative, grant number (RGP.2/469/45). The funders had no role in study design, data collection and analysis, decision to publish, or preparation of the manuscript.

### Grant Disclosures

The following grant information was disclosed by the authors:
The Deanship of Scientific Research and Graduate Studies at King Khalid University in Abha, Saudi Arabia, through the Large Research Group initiative: RGP.2/469/45.

### Competing Interests

Ajinkya M. Pawar serves as an Academic Editor for PeerJ.

### Author Contributions

- Kalyani Garde conceived and designed the experiments, performed the experiments, analyzed the data, prepared figures and/or tables, and approved the final draft.
- Ajinkya M. Pawar conceived and designed the experiments, performed the experiments, analyzed the data, authored or reviewed drafts of the article, and approved the final draft.
- Anuj Bhardwaj conceived and designed the experiments, performed the experiments, authored or reviewed drafts of the article, and approved the final draft.
- Jatin Atram conceived and designed the experiments, performed the experiments, prepared figures and/or tables, and approved the final draft.
- Suraj Arora conceived and designed the experiments, prepared figures and/or tables, and approved the final draft.
- Dian Agustin Wahjuningrum conceived and designed the experiments, authored or reviewed drafts of the article, and approved the final draft.
- Maria Febritania Wahyuni Huri conceived and designed the experiments, prepared figures and/or tables, and approved the final draft.
- Dennys Kurnia conceived and designed the experiments, prepared figures and/or tables, and approved the final draft.

### Human Ethics

The following information was supplied relating to ethical approvals (i.e., approving body and any reference numbers):

The Institutional Ethics Committee of the College of Dental Science & Hospital (CDSH/740/2023; date of approval 01-08-2023).

### Ethics

The following information was supplied relating to ethical approvals (i.e., approving body and any reference numbers):

The Institutional Ethics Committee of the College of Dental Science & Hospital (CDSH/740/2023; date of approval 01-08-2023).

## Data Availability

The raw data of the responses are available in the Supplemental File.

## Supplemental Information

Supplemental information for this article can be found online at http://dx.doi.org/10.7717/peerj.19455#supplemental-information.

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
