# Peer review of "Knowledge, perceptions, and practices of dental professionals and students regarding obturation in endodontic procedures: a nationwide cross-sectional survey"

_PeerJ, doi:10.7717/peerj.19455_

## Round 0.1 · original submission · Major Revisions

Dear authors, thank you for your submission.

After reviewing your study, several key concerns were raised that need to be addressed before publication.

One reviewer highlighted issues with the survey design, noting that some questions and answer choices were not fully appropriate for assessing the knowledge of dental practitioners and students. It is recommended that the questionnaire be simplified and revised to focus on relevant and clear questions. Not sure this could be done at this time, but certainly some reasoning for the methodology used should be clarified; as well as potential limitations. Another concern involves the heterogeneity of the participant group, which includes both dental students and graduated dentists. The differences in experience between these groups could impact the reliability of the results. It was suggest either focusing on one group or addressing how this variability may influence your findings, particularly if you choose to retain both groups. Another option is to simply analyze both groups separately and potentially compare them. Please, refer to the reviewers' comments for further details. Looking forward to receiving your revised submission.

Reviewer 1 ·

Basic reporting

This study tackles a critical topic, offering valuable insights into obturation processes in endodontic treatments. With 422 participants, it establishes a robust foundation for reliable and generalizable findings. The use of a structured questionnaire and appropriate statistical methods, such as the Chi-Square test, ensures the research questions are effectively addressed. Additionally, the study highlights key findings that could drive improvements in material selection and obturation techniques. Notably, the participants' openness to adopting new technologies and materials further underscores the study's relevance and forward-looking approach. However there are some areas that need to be improved.

Experimental design

Methodology Section:
• The process of developing the questionnaire should be described in greater detail (e.g., how were validity and reliability assessed?).
• The use of "snowball sampling" for participant recruitment may introduce potential biases; measures taken to mitigate these should be explained.

Validity of the findings

Presentation of Results:
• While the high utilization rate of lateral compaction is highlighted, the reasons for less frequent use of other techniques are not discussed.
• The reasons behind retreatment cases and their impact on patient outcomes should be elaborated further.
Discussion Section:
• The findings should be more comprehensively compared with similar studies in the literature.
• The low usage of bioceramic sealer could be discussed in the context of technology adoption.

Additional comments

Figures and Tables:
• Tables and figures could be made more user-friendly and summarizing. For example, graphical representation of findings would facilitate quicker understanding by readers.
References:
• Some references could be updated or supplemented with more recent literature. Emphasis should particularly be placed on studies from the past five years.

Reviewer 2 ·

Basic reporting

no comment

Experimental design

no comment

Validity of the findings

no comment

Additional comments

I appreciate the authors for their effort to perform such a comprehensive study. However, after careful review, I noticed some issues, particularly related to the formulation of the questionnaire. I would like to comment on these issues as follows;
1. In line 61, the word "intrusive" should be replaced with "invasive". The authors should revise the manuscript in terms of English language.
2. In line 86, the phrase "single-cone obturation" should be deleted. It is irrelevant.
3. The null hypothesis was not written.
4. In line 211 “The majority of 254 individuals (60.2%) prioritized moisture resistance, bactericidal capabilities, and non-staining to tooth structure, while 48 people (11.4%) emphasized sealing ability, biocompatibility, and dimensional stability as important attributes (p<0.05) (Fig. 2).”
These findings are inconsistent with Figure 2 and Question 4 in the knowledge section of the questionnaire. The question does not include choices for “Moisture resistance, bactericidal capabilities, and non-staining to tooth structure.”
5. “Which material is commonly used for obturation in endodontics?”
The answer choices in this question are irrelevant. Currently, amalgam or composite materials are not used for root canal fillings. Instead, MTA or bioceramic materials could be considered as answer choices. Amalgam or composite might be used as retrograde filling materials in apical surgery, not root canal treatment.
Camilleri, J. (2017). Will bioceramics be the future root canal filling materials?. Current Oral Health Reports, 4, 228-238.
6. Some of the questions and choices in the survey are not sensible for assessing the knowledge of dental practitioners. For instance, asking whether they use amalgam or composite fillings does not make sense, as there is no mention in the endodontic literature of using amalgam or composites as root canal filling materials. Silver cones were utilized in the past, but nowadays it is almost impossible to find silver cones on the market. I recommend authors reduce the number of questions and simplify the questionnaire, focusing only on questions with rational and relevant choices.
7. the answer choices of question 8 are irrelevant in the knowledge section. One possible consequence of inadequate filling is a persistent infection, which may lead to periapical lesions.
Liang, Y. H., Li, G., Shemesh, H., Wesselink, P. R., & Wu, M. K. (2012). The association between complete absence of post-treatment periapical lesion and quality of root canal filling. Clinical oral investigations, 16, 1619-1626.
8. in the attitude section, in the first question, before asking about the confidence level regarding canal obturation, it is essential to inquire about the technique used by the participant. This is important because confidence levels can vary depending on the obturation technique employed in endodontic treatment. Also, the practitioner's experience level impacts the confidence level. Therefore, authors should have included only students or graduated dentists in their study.
Kucuk, M., Ratakonda, M., & Piasecki, L. (2024). Teaching a new obturation technique in preclinical endodontic training: assessment of student learning experience, performance, and self-evaluation. Journal of Endodontics, 50(11), 1634-1641.
9. Answers to Question 7 from the attitude section and Question 3 from the practice section may vary based on the obturation technique and the material used. These factors should have been considered prior to asking question 7 and 3.
10. I suggest authors analyze these responses concerning the obturation technique employed and the participants' experience levels. Private practitioners may exhibit greater confidence and encounter fewer challenges than third-year students.
11. “Are you satisfied with the current obturation techniques you use in terms of efficiency and predictability?
Analyzing the answers of this question is not appropriate, as the answers depend on the obturation techniques employed by practitioners. In the literature, warm vertical compaction is generally reported to be more challenging than the single cone obturation technique. Also, hydraulic condensation is a simple and efficient obturation technique.
AlBakhakh, Bahaa, et al. "Rapid Apical Healing with Simple Obturation Technique in Response to a Calcium Silicate‐Based Filling Material." International Journal of Dentistry 2022.1 (2022): 6958135.
12. In line 266, the word "relentlessly" could be replaced with "regularly" for clarity.
13. “Do you actively seek feedback from patients regarding their experience with root canal obturation?”
Analyzing the responses to this question does not have scientific significance. Dentists usually seek feedback regarding the outcomes of root canal treatments. If the authors had investigated the specific obturation technique such as hydraulic condensation in this study, asking this question might have been more reasonable.
14. In line 371, “Only 11.1% of participants were unhappy with their current efficiency and predictability of obturation processes, indicating that there is opportunity for development and innovation in the industry.”
With the invention of bioceramic sealers, hydraulic condensation became so popular among practitioners across the world due to its simplicity and favorable treatment outcomes. The authors state that 11.1 % of participants are unhappy with current obturation techniques and the majority (69.7%) believe that improvements are needed in the current obturation techniques. The authors interpreted this finding that there is an opportunity for the development of innovation in the industry. However, using hydraulic condensation with the bioceramic sealer is a huge novelty. The participant might have not had the opportunity to access these materials and use them to report their experience. Because bioceramic sealer could be expensive in developing countries. 40% of Participants already reported that only 27.3% used bioceramic sealer in routine practice.
15. In line 376, “The disparity in Bioceramic Sealer acceptability (11.1%) instead of the substantial percentage of people not utilizing it (41.0%) highlights the need for increased education and knowledge regarding emerging obturation materials like Bioceramic Sealer.”
The low usage frequency of the bioceramic sealer was linked to the education level of the participants. However, the high cost of the bioceramic sealer may also explain why many practitioners choose not to use it. Traditional sealers are more affordable than bioceramic sealers.
Aminoshariae, Anita, Carolyn Primus, and James C. Kulild. "Tricalcium silicate cement sealers: Do the potential benefits of bioactivity justify the drawbacks?." The Journal of the American Dental Association 153.8 (2022): 750-760.

Reviewer 3 ·

Basic reporting

The article is written in English, using simple, professional language. However, the whole document requires editing.
Some statements need to be improved upon, such as in lines 71, 103-104, 163-164, while some words should be checked e.g in-tern (line 156), "com-bination" (line 28)
Adequate literature references have been provided. The background and context of obturation were well documented, but more details regarding advances in the obturation techniques are needed.
The manuscript has been structured into standard sections: “Introduction,” “Methodology,” “Result,” “Discussion,” and “Conclusion.”
The figures are relevant to the content, but I'm unsure whether the resolutions are sufficient. The figures were appropriately described and labelled, but figures 2-10 had so many questions put together as a figure of many bar charts/histograms. Maybe some might have been better displayed as a table.
The appropriate data were provided in the figures.
The submission includes all results relevant to the hypothesis, but some extra results were added, e.g., lines 262-269.

Experimental design

This was an original primary observational research with use of a developed validated survey questionnaire. Ethical considerations were well taken.

The justification of the study was not clear

Validity of the findings

In the methodology, the authors explained how the research tool (questionnaire) was validated among 42 participants but it was not clear if these participants were part of the main survey.
The author should give more details of the different sections of the questionnaire for easy modification and replication.

The conclusions were in line with the result provided.

Additional comments

Title: The authors need to consider another title reflecting what was done. The research title specifies that the survey was about dental students. However, the body of the manuscript revealed that the study was conducted amongst students (2 grades, which comprised only 18%. Fig 1), interns (42%), private dental practitioners, and dental faculties.
Introduction: The authors need to provide more details on advances in obturation materials and techniques that are known and expected to be known. Furthermore, the justification was not strong. The authors should give a strong argument on the need for the study.
Methodology.
Sampling size: How did the authors arrive at a sample size of 422? The sampling size calculation should be revisited. The 20% attrition of 345(calculated sample) is 69; this added will give 414.

Sampling technique: The snowballing technique is still being queried in this study. According to a statement, 422 survey questionnaires were sent out by the researchers, but it was also mentioned that they (the authors requested that some participants helped spread to colleagues). More clarifications should be provided by the authors in this regard.
The authors need to consider the effect of including all the cadres of professionals included in the survey. The respondents' levels of experience are wide apart, as they vary greatly in experience, knowledge, skill, etc.
The study's objective was not well defined. Although knowledge, attitude, and practice were included in the title, many other things were reported based on the survey questions. The authors should maybe just limit the report to these areas.

The explanation on the content of the survey questionnaire should be more detailed.
Data analysis: The authors should try to define the data analysis. Descriptive data was well taken, but the inferential analysis should have been more detailed. What variables did the authors compare for statistically significant gender comparison?? (Line 193-194)
Result
The result was presented in a systematic way along the lines of knowledge, attitude, and practice. However, alongside the presentation of the result, the authors gave an explanation (from lines 251 to 280 under Practice). This should be reviewed. The explanation for the result should be discussed.

Discussion:
The authors need to clarify the discussion. Some statements contradicted the result. For Line 364, the lateral compaction technique should be checked.
Overall, the discussion should be tailored to the result.

Annotated reviews are not available for download in order to protect the identity of reviewers who chose to remain anonymous.

---

## Round 0.2 · Major Revisions

Dear Authors,

Thank you for your continued efforts to improve the manuscript. After reviewing the latest revisions and the feedback from the reviewers, I believe that several aspects can and should be better addressed and improved.

One reviewer has raised concerns regarding the validity and reliability of the findings, and while these concerns are valid, I do not believe they warrant an outright rejection. As such, I have decided to recommend major revisions instead.

Here are the key points I suggest you address in your revision:

- Participant Diversity and Experience Levels:
One reviewer raised the issue of having participants with varying levels of experience in endodontics (students, interns, postgraduates, and professionals). While this may create some variability, you can justify the inclusion of a broad range of participants by explaining the goal of obtaining a comprehensive understanding of the knowledge and practices across different experience levels and that it reflects real-world practices and provides a more holistic understanding of the knowledge and practices surrounding endodontic obturation; clearly and in the manuscript, not just in the rebuttal.
To further address this concern, I recommend conducting a stratified analysis of the data, where you present results separately for different groups (e.g., students, interns, professionals). This would help clarify how experience levels influence the findings and ensure that the data are not skewed by any particular group. Additionally, a more focused analysis of relevant observations will strengthen the conclusions drawn from the data.

- Inclusion of Outdated Materials (e.g., Silver Cones, Amalgam):
The reviewer questioned the relevance of including outdated materials like silver cones and amalgam in the survey. However, i do recognise that this may be out-of-touch with different global geopolitical contexts. While these materials may no longer be widely used, it is important to acknowledge that dental practices vary globally, and some regions or practitioners may still use them.
I recommend justifying the inclusion of these materials by explaining that the study aimed not only to assess current practices but also to capture perceptions and knowledge about older materials; what is known or perceived by dental professionals. Some participants may still be familiar with or use older materials, and this could be valuable information for understanding gaps in education or outdated practices. The authors might also consider revising the manuscript to highlight the prevalence of contemporary materials (e.g., bioceramics, gutta-percha) in their analysis, so the focus remains on current trends. I know that this has been tried, but a better revision of this aspect could help identify gaps in education or highlight outdated practices still in use. I suggest revising the manuscript to emphasize contemporary materials (such as bioceramics and gutta-percha) in your analysis, ensuring that the focus remains on local context of current trends while acknowledging older materials for context.

- Context for Confidence-Related Questions:
The reviewer pointed out that certain questions, such as those related to confidence in obturation, lack context regarding the specific techniques used by participants. This could affect the reliability of the responses.
While revising the questionnaire at this stage may not be feasible, I recommend adding a section in the manuscript to address the potential variability in responses due to different techniques. Ie the authors could also amend the manuscript to explain how they accounted for potential variability in responses due to different techniques. For example, they might introduce additional analyses that account for technique-specific confidence levels. If technique-specific data is not available, this limitation should be very well acknowledged in the manuscript. Additionally, you may want to revise that future research explore the influence of technique on confidence in obturation, as this could be an important area for further investigation.

- Focus on a Specific Group:
Another point raised was the suggestion to focus the study on a more specific group, such as students or recent graduates, to enhance the reliability of the findings. While narrowing the focus to a homogeneous group could improve the analysis, I believe your decision to include a broader range of participants is justifiable, as it reflects real-world dental practices where professionals of varying experience levels interact with different obturation techniques and materials.
To strengthen your manuscript, consider conducting subgroup analyses to show how experience levels influence the results. You can also emphasize the generalizability of your findings, given the diversity of your sample. It is important to highlight this aspect in your revised manuscript.

- Irrelevant Findings:
The reviewer suggested focusing the manuscript more narrowly on the study’s objectives and excluding irrelevant findings. I recommend that you review the findings and ensure that the manuscript clearly emphasizes those that are most relevant to your central questions. If you decide to retain the broader scope of your study, ensure that the relevance of these findings to the larger objectives is clearly explained in the manuscript. It may be beneficial to downplay less relevant findings in the discussion, but I do not recommend removing them entirely, as they may still provide useful context.

In conclusion, I believe that addressing these concerns will significantly enhance the scientific rigor of your study and improve its overall impact.

Reviewer 1 ·

Basic reporting

No comment

Experimental design

No comment

Validity of the findings

No comment

Additional comments

No comment

Reviewer 2 ·

Basic reporting

no comment

Experimental design

no comment

Validity of the findings

no comment

Additional comments

This manuscript explores an interesting topic: the knowledge, perceptions, and practices of dental professionals regarding root canal obturation. I read the manuscript twice, carefully looking into the questions and results step by step.

However, I identified several issues with the survey design that raise concerns about the reliability of the results, leading me to recommend rejecting this manuscript. Below, I added the specific areas that influenced my decision:

1. The participants in this study have varying levels of experience in endodontics, which could result in unreliable data. The researchers should have focused on a specific group, such as students, recent graduates, or specialists, when analyzing the data.

2. Some questions are not relevant for obtaining answers or conducting meaningful analysis, as certain materials listed are not commonly used in modern obturation practices, despite their presence in the market. For instance, options such as silver cones and amalgam are outdated. A wide range of contemporary materials should have been included as choices.

3. The formulation of some questions is inappropriate because additional context is needed before they can be answered accurately. For example, in the attitude section, the first question regarding confidence levels in canal obturation should first ask about the technique used by the participant. This is important because confidence levels can vary based on the obturation technique employed in endodontic treatment. Additionally, a participant's level of experience affects their confidence. Therefore, the authors should have focused exclusively on students or graduates in their study.

Reviewer 3 ·

Basic reporting

The authors have been able to address all the queries raised.
The document reads better. However, some minor corrections need to be made. (see manuscript)

Experimental design

No Comment

Validity of the findings

No comment. All queries addressed.

Additional comments

The manuscript has significantly improved. However, as pointed out in the earlier review, the authors should try to focus on the study's objectives and leave out some irrelevant findings to the study.

Annotated reviews are not available for download in order to protect the identity of reviewers who chose to remain anonymous.

---

## Round 0.3 · accepted · Accept

Dear authors, i am now accepting this work for publication. Congratulations! Please, be throughout with your proofreading before the final publication.